# A Physiologically Based Pharmacokinetic Framework for Quantifying Antibody Distribution Gradients from Tumors to Tumor-Draining Lymph Nodes

**DOI:** 10.3390/antib11020028

**Published:** 2022-04-14

**Authors:** Eric Salgado, Yanguang Cao

**Affiliations:** 1Department of Pharmacotherapy and Experimental Therapeutics, Eshelman School of Pharmacy, University of North Carolina at Chapel Hill, Chapel Hill, NC 27599, USA; esalgado@email.unc.edu; 2Lineberger Comprehensive Cancer Center, University of North Carolina at Chapel Hill, Chapel Hill, NC 27599, USA

**Keywords:** therapeutic antibodies, pharmacokinetics, immunotherapy, PBPK, neoadjuvant, tumor-draining lymph nodes

## Abstract

Immune checkpoint blockades prescribed in the neoadjuvant setting are now under active investigation for many types of tumors, and many have shown early success. The primary tumor (PT) and tumor-draining lymph node (TDLN) immune factors, along with adequate therapeutic antibody distributions to the PT and TDLN, are critical for optimal immune activation and anti-tumor efficacy in neoadjuvant immunotherapy. However, it remains largely unknown how much of the antibody can be distributed into the PT-TDLN axis at different clinical scenarios. The goal of the current work is to build a physiologically based pharmacokinetic (PBPK) model framework capable of characterizing antibody distribution gradients in the PT-TDLN axis across various clinical and pathophysiological scenarios. The model was calibrated using clinical data from immuno-PET antibody-imaging studies quantifying antibody pharmacokinetics (PK) in the blood, PTs, and TDLNs. The effects of metastatic lesion location, tumor-induced compression, and inflammation, as well as surgery, on antibody concentration gradients in the PT-TDLN axis were characterized. The PBPK model serves as a valuable tool to predict antibody exposures in various types of tumors, metastases, and the associated lymph node, supporting effective immunotherapy.

## 1. Introduction

The success of antibody immunotherapy in advanced metastatic cancers has inspired oncologists to assess these therapies in patients with early-stage cancers or in the neoadjuvant setting [1]. Early results for neoadjuvant immunotherapy are promising. In a metastatic melanoma study, about 30% of patients showed a significant or complete pathological response after a single neoadjuvant dose of anti-PD-1 antibody pembrolizumab [2]. Significant pathological responses were also observed in non-small-cell lung cancers (NSCLC) with neoadjuvant nivolumab and ipilimumab combination therapy, owing to the activated, systemic T-cell immunologic response [3]. The primary tumor (PT) and tumor-draining lymph node (TDLN) signaling axis has been a focal point for the efficacy of neoadjuvant immunotherapy [1].

An intact PT–TDLN signaling axis is critical for the efficacy of immunotherapy, as it may enhance tumor-specific T-cell priming and expansion at either the primary tumor microenvironment or in the neighboring TDLNs. The concept of “Cancer-Immunity Cycle” underlines the importance of an intact PT-TDLN signaling axis for systemic antitumor immunity [4]. The full activation of antitumor immunity is also highly dependent on adequate therapeutic antibody exposure into both the primary tumor and TDLNs. Unfortunately, surgical resections or other pathological conditions may disrupt the tissue anatomical structure and impair the effective distribution of therapeutic antibodies into the tumor lesions and the surrounding TDLNs, leading to a suboptimal effect of immunotherapy.

Moreover, checkpoint blockade immunotherapies reveal organ-specific patterns of response. The lesions in the lymph nodes tend to be the most responsive site for systemic activation of tumor-specific T cells [5]. The activation of tumor-specific T cells in the TDLNs is enhanced by sufficient antibody exposure. However, it remains largely unknown how much antibody can distribute into the PT–TDLN axis at different anatomical sites and clinical scenarios. Therefore, an exploration of the processes that affect the antibody distribution kinetics and the resulting degree of target engagement in the organ-specific PT–TDLN axis has become critical for understanding the mechanisms contributing to the efficacy of neoadjuvant immunotherapy.

Physiologically based pharmacokinetic (PBPK) modeling allows for the quantitative analysis and prediction of drug pharmacokinetics (PK) and tissue distributions (i.e., concentration vs. time profiles) at any relevant organ. In PBPK models, physiologically relevant parameters, such as organ-specific lymph flows and lymphatic networks, can be implemented to support anatomical characterizations of antibody distribution [6]. PBPK models could define antibody distribution gradients along the organ-specific PT–TDLN axis. While there are previously reported antibody PBPK models, none to date have emphasized antibody biodistributions in anatomically distinct tumors and their TDLNs [7,8]. We have reviewed the lymphatic network and the general framework for building such a model [6]. The current work provides the first proof-of-concept PBPK model to characterize antibody distribution gradients from organ-specific tumor to TDLNs. The PBPK model serves as a valuable tool to predict antibody distribution gradients between the tumor and TDLN in a spatiotemporal manner, which has substantial implications for neoadjuvant immunotherapy.

## 2. Materials and Methods

### 2.1. The PBPK Model Structure

The proposed PBPK model structure is shown in Figure 1**,** adapted from our minimal PBPK model with emphasis on the anatomically distinctive tumors and the associated TDLNs [6,9]. In the model, all non-tumor tissues were lumped together as one compartment to account for the antibody distribution in the other parts of the body. Antibody systemic clearance from the vascular compartment (CL_p_) is assumed to be the primary nonspecific elimination pathway for therapeutic antibodies, consistent with previous findings [10]. Moreover, while TDLNs are vascularized by both blood and lymphatic vessels, antibody delivery to TDLN was only assumed via lymphatic capillaries. Specialized, thickened endothelial cells in the LN vasculature known as high endothelial venules (HEVs) facilitate active transport of lymphocytes from the blood into the LN via adhesion receptors; due to their structure and unique function, it was assumed that direct antibody entry via convection was negligible (σ_V,TDLN_ = 1) [11,12]. The target-mediated antibody accumulation and elimination were considered in both the tumors and TDLNs. The target-binding process is assumed to be at a quasi-equilibrium state, where the process of target association and dissociation were not separately considered. This assumption has been widely validated in many cases considering that antibody-target binding is usually on a much faster time scale than those describing target synthesis (k_syn_), degradation (k_deg_), and internalization (k_int_) [13,14]. Furthermore, we are not taking into accounting residualized/metabolized antibodies that may increase noise signal. Antibody trafficking from the tumors to the TDLN, determined by the physical distance and lymphatic network, are subject to non-specific pinocytosis and FcRn salvage in the endothelial cells, while trafficking through the lymphatic vessels. FcRn expression levels in lymphatic endothelial cells were assumed to be the same as for blood vascular endothelial cells. A detailed summary of the model ordinary differential equations is provided in Appendix A.

### 2.2. Model Calibration and Simulation

Antibody human immuno-PET data in the blood, tumors, and TDLN from eleven IgG antibodies were applied to calibrate the PBPK model. We applied the WebPlotDigitzer tool [15] for digitizing the disposition data of the eleven antibodies across eight immuno-PET biodistribution studies. Immuno-PET imaging studies involve the radiolabeling of antibodies for quantitative in vivo tracking of antibody binding in the PT and TDLN. The eleven unique immuno-PET antibody:primary tumor pairs included: ^89^Zr-fresolimumab (glioblastoma) [16]; ^89^Zr-bevacizumab (non-small-cell lung cancer; NSCLC) [17]; ^89^Zr-bevacizumab (breast cancer) [18]; ^89^Zr-bevacizumab (renal cell carcinoma) [19]; ^64^Cu-DOTA-trastuzumab (breast cancer) [20]; ^89^Zr-trastuzumab (esophagogastric cancer) [21]; ^89^Zr-MMOT0530A (pancreatic cancer) [22]; ^89^Zr-MMOT0530A (ovarian cancer) [22]; ^89^Zr-atezolizumab (NSCLC) [23]; ^89^Zr-atezolizumab (breast cancer) [23]; ^89^Zr-atezolizumab (bladder cancer) [23]. Of note, the therapeutic antibodies used for model calibration exhibit similar distribution behaviors to most immunotherapeutic antibodies and these antibodies are all approved for treating various tumor types. Whenever there is data available for calibration from a study, they are presented as the mean +/− standard deviation (SD). Curves that are not calibrated with any literature data (i.e., no data points in curve) represent what the predicted (simulated) SUV would be at that tissue. For clarity and easier viewing, the SD error bars are only presented in one direction.

We used the RxODE simulation package in R to build the model (Appendix A) and perform model calibration. Model parameters were derived either from the literature, theoretical calculation, or model optimization. Once calibrated, the PBPK model was subsequently applied to simulate and characterize antibody distribution gradients from organ-specific tumors to TDLN across three clinical scenarios: metastasis, inflamed tumor microenvironment, and before and after surgical resection (i.e., neoadjuvant vs. adjuvant). Sensitivity analyses were also performed to assess the parameters that significantly influence antibody distribution. The key parameters (excluding the vascular and lymphatic reflection coefficients) were simulated in the range of 0.1–100 fold of the optimized values. The σ_V_ was assessed in the range of 0.01–1.25, and σ_L_ in the range of 0.1–5 times the optimized values. The fold change values were selected such that they reflect a wide, yet reasonable spectrum (i.e., σ_V_ and σ_L_ are restricted to values between 0 and 1) of plausible values for these parameters and to assess their impact on model predictions.

## 3. Results

### 3.1. The PBPK Model Adequately Captured Antibody Distribution in Anatomically Distinctive Tumors and TDLNs

The developed PBPK model adequately captured antibody distributions across multiple types of anatomically distinctive tumors and TDLNs (Figure 2). Table 1 and Table 2 denote the optimized range of parameters for calibrating each organ-specific tumor model. The parameter ranges are associated with tumor-specific and antibody-specific properties (i.e., σ_V,_ R01, R02, Kd, dln, CL_p_), highlighting the variabilities in antibody distribution across tumors. For instance, renal tumors showed the highest antibody uptake (i.e., highest SUV value), which was about five-fold higher than antibody uptake in the brain and lung cancers. This is associated with the kidney being one of the most highly perfused organs in the body. Indeed, our model is consistent with clinical observations that the kidney tumors also exhibit the fastest blood and subsequently lymph perfusion rates [24]. Among the tumor types with both PT and TDLN data, the kidney model also exhibited the largest distribution gradient between the two tissues. Our calibrated parameters suggest that the very high tumor target density of the PT (R01) compared to TDLN (R02) accounts for this steep antibody gradient. In the other tumor types where the observed distribution gradient was relatively flat, the tumor target density between these two tissues were comparable. The reasoning behind this clinical observation is discussed in-depth in the following section.

In addition to metastatic lesions, other clinical and pathophysiological processes, such as surgery and rapid tumor growth-induced inflammation, may further alter the antibody distribution gradient from the PT to TDLN. Simulations to capture these processes were thus performed using the calibrated model, and their impacts on the distribution gradients were also compared.

### 3.2. Therapeutic Antibodies Showed Varied Distribution across Metastatic Lesions

Lung metastasis data in patients with both bladder and breast cancer primary tumor types were compared in the modeling platform [23]. The PBPK model could capture antibody distribution in lung metastases in patients with differing tumor origins (Figure 3a,b). In bladder cancer, antibody distribution into the TDLNs was lower than the primary tumors, but higher than the metastatic lesions (Figure 3a). In breast cancer, antibody distribution into the TDLNs was higher than both the primary tumor and metastases (Figure 3b). Antibody distribution gradients across metastases were also compared at either comparable (Figure 3c) or different simulated target density (Figure 3d). The antibody distribution gradient from the newly metastasized lesion to its TDLNs are dependent on various factors: target density (R01, R02), tumor vascular leakiness (σ_V_), and lesion locations (dln, L_organ_). Target density at either the PT (R01) or TDLN (R02) can significantly alter the antibody distribution gradient between these two tissues, especially at low antibody concentrations in the tumors relative to target levels. At a low antibody concentration, high target abundance and extensive target-mediated antibody endocytosis can make antibodies quickly degrade in the tumors, further lowering antibody exposure in the tumor beds [39]. Therefore, inside a target-expressing metastatic lesion (high R01), the exposure of free antibody (i.e., Cf1) left for trafficking into the TDLN will be limited, causing a sharp distribution gradient, as shown in Figure 3d.

In addition to the target density in metastatic lesions, the metastatic sites also affect the antibody distribution gradient from the PT to TDLN. Differences in organ-specific lymph flow and the physical distance between the organ-specific PT and TDLN (Table 1 and Table 2) influence both the rate of antibody distribution and the extent of FcRn recycling along with the organ-specific lymphatic vessel network, leading to organ-specific distribution gradients. The organ-specific distribution gradient was also observed for albumin from the injection site to the sentinel lymph nodes [34]. The antibody distribution gradient in the TDLN at different anatomical sites partly explains the heterogeneous responses across metastatic lesions [5]. The results from Table 2 suggest that antibodies with similar binding affinities to FcRn (i.e., Kd_FcRn_) will be subject to a similar distribution gradient (i.e., dln), with the steepness of the gradient governed by the length of the lymphatic vessel network that drains that region.

### 3.3. Therapeutic Antibodies Markedly Reduced Distribution in the TDLNs after Surgical Resection

The effect of surgical resection on antibody distribution gradients was investigated next using the PBPK model and parameterizations (Figure 4 and Appendix D). Surgical resection of the primary tumor (i.e., R01 = 0) resulted in the impairment of local lymphatic drainage surrounding the tumor region, clinically known as lymphedema [40]. Given that the degree of surgery-induced lymphedema varies widely, we simulated the effect of both a 50 and 80% reduction in local lymph flows (i.e., L_organ_, L_aff_) on the antibody distribution gradient. Consistent with the expected clinical outcomes, our simulation suggested a significantly reduced distribution to the residual TDLNs, which may result in the suboptimal exposure of the antibody. Inefficient T-cell priming in the residual TDLNs may partially explain the heterogeneous responses across patients in adjuvant settings. Therefore, it could be challenging to treat the residual lymph node-positive patients with immunotherapy after surgical resection of the tumors due to poor drug distribution. This observation has significant implications for treating minimal residual disease and the relapse potential during adjuvant immunotherapy.

### 3.4. Tumor-Induced Tissue Inflammation Limits Antibody Distribution in the Intratumoral TDLNs

We then investigated the effects of rapid tumor growth-induced inflammation on antibody distribution gradients across several TDLN networks (Figure 5 and Appendix E). Due to rapid tumor expansion, the intratumoral lymphatic vessels and the lymphatic drainage to the intratumoral TDLNs often become compressed, rendering a non-functional lymphatic system (i.e., L_aff_~0) [41]. Subsequently, the lymphatic flow to the peri-tumoral TDLNs is considerably increased in the rapidly growing tumor [42]. The increased afferent lymph flow (L_aff_) and their impact on antibody distribution gradient were investigated in the model. As shown in Figure 5, our simulations suggest that the antibody distribution gradient between various TDLN networks could be drastically altered because of the rapid expansion of the tumor and tumor-induced inflammation. The antibody distribution to the intratumoral TDLNs is considerably lower than to the peritumoral TDLNs, regardless of TDLN tumor metastatic status. The impaired antibody distribution into the intratumoral TDLNs could result in a sub-optimal antibody exposure for poor anti-tumor response in the altered TDLN networks.

### 3.5. Sensitivity Analysis

Finally, a series of local sensitivity analyses were performed to identify the key model parameters that most highly impact antibody distribution gradients from the tumors to the TDLNs. As shown in Figure 6 and Appendix C, nearly all parameters were shown to affect, to varying degrees, antibody distribution profiles. Furthermore, as the parameters R01/R02 and dln were increased, the simulated profiles also increased (i.e., shifted upward), whereas parameters K_d_ and k_int_ showed an inverse simulated effect when increased. k_deg_ was not a sensitive model parameter. These trends are largely expected, given that having a relative higher PT target density (R01/R02) should lead to higher antibody exposure in the tumors. Moreover, a slower (i.e., lower) antibody internalization rate (k_int_) along with lower antibody K_d_ values will both lead to prolonged antibody-target engagement and subsequently higher antibody exposure in tumor beds. Future clinical studies with richer data would allow for a more thorough calibration of the model parameters.

## 4. Discussion

The high potential for prescribing immunotherapy in neoadjuvant settings has made it critical to understand therapeutic antibody distribution in the tumor and the TDLNs. Despite previously published PBPK models for quantifying antibody PK in tumors and the lumped LNs, none have characterized the organ-specific antibody distribution gradient from the PT to the TDLNs [7,8]. The developed proof-of-concept PBPK model could thus be valuable to predict the antibody distribution gradient between these two tissues in an anatomically distinctive and pathophysiologically relevant manner. The model was calibrated using data collected in various types of cancer patients in immuno-PET antibody studies. The calibrated PBPK model was used to simulate the antibody distribution gradient between the tumors and TDLNs at different pathophysiological and clinical scenarios, including metastasis, surgical resection, rapid tumor expansion, and expansion-induced inflammation.

The developed PBPK framework predicts antibody distribution largely based on the available tumor-specific information, including tumor type, anatomical location, tumor target density, target abundance, surrounding lymphatic network, and surgical resection. Although broad validation is needed, the model could provide a population-average prediction of antibody exposure in the tumors and the TDLNs. Once patient-specific information becomes available, the model could be further improved with high precision. Of note, the priori exposure prediction could be informative to comprehending therapeutic efficacy, resistance probability, and dose justification.

Most checkpoint blockades are approved for treating advanced cancers when multiple metastases have developed. Metastasis is arguably the most significant concern in cancer treatment, accounting for over 90% of cancer deaths [43]. High metastasis and target-expressing cells are usually associated with rapid tumor antigen-mediated antibody degradation, resulting in a steeper distribution gradient and limited antibody exposure to the TDLNs (Figure 3d). Indeed, an increased trastuzumab clearance was reported in breast cancer patients with high tumor target density due to increased antigen-mediated antibody clearance [44]. Furthermore, the metastatic location of the tumor may also affect the concentration gradient to the TDLN. Lymphatic drainage networks are quite heterogeneous across anatomical sites [6], differing in the length and networks of the lymphatic vessels around tumors to the TDLNs [34]. As our model predicted, the trafficking distance from the tumor to the TDLNs can be associated with the fraction of antibody degradation in the lymphatic endothelial cells, even though FcRn can recycle a fraction of antibodies.

Surgical resection of the primary tumor was also evaluated, which was predicted to substantially affect antibody distribution into the TDLNs, potentially leading to sub-optimal antibody exposure (Figure 4). As shown in Figure 5, heterogeneous distribution of the antibody was observed in both the peri-tumoral and intra-tumoral lymphatic vessels and the ancillary TDLNs, highlighting the impact of rapid tumor expansion and expansion-induced inflammation on antibody distribution to various TDLN networks. The heterogeneous and sometimes inadequate antibody exposure to tumor metastatic lesions at varying anatomical sites may create a tumor sanctuary, leading to rapid tumor relapse. Our model platform once validated with more extensive data would predict the metastatic lesions where antibodies have limited exposure and tumor cells could possibly relapse.

One limitation in our model is the scarcity of the data for model calibrations. Few studies quantified antibody uptake in the TDLNs, and the longitudinal data were not sufficient to optimize all dynamic model parameters. Information about the tumor target lesions and the specific anatomical locations of the TDLN were not accurately reported. The parameters with high uncertainty were restricted to the literature values. Notably, despite similar PK and distribution profiles, some antibodies used in model calibration were not checkpoint blockades. While the afferent and efferent lymph flows for each organ-specific tumor were fixed to the same value in this study, it is plausible they could vary, considering the different densities and properties of lymphatic vessels across tumor types. Furthermore, an empirical approach was used to account for FcRn-mediated salvage occurring in the lymphatic vessels (Appendix B). Indeed, there are already several published PBPK models that adequately capture these processes in detail and were not the focus of this study [27,29,45]. Moreover, depending on the study, the SUV value reported and used for model calibration was either SUVmean, SUVmax, or SUVpeak. SUV values are generally viewed as a semiquantitative imaging metric due to the uncertainties around its measurement (i.e., effect of residualizing radioisotopes, high residual variability, etc.) [46]. This also explains the high variabilities seen in the observed data (i.e., high standard deviations). Therefore, the model should be used as more of a qualitative tool to predict the distribution gradients in relevant scenarios.

We performed a series of local sensitivity analyses to assess the robustness of the model and highlighted the parameters significantly influencing antibody distribution gradients in the TDLNs. The most sensitive model parameters should be further optimized in future studies before making patient-specific predictions.

## 5. Conclusions

In summary, the current PBPK model presented a proof-of-concept quantitative platform that can predict antibody distribution gradients from the tumors to the TDLNs in a spatiotemporal manner, which has substantial implications for neoadjuvant immunotherapy.

## Figures and Tables

**Figure 1 antibodies-11-00028-f001:**
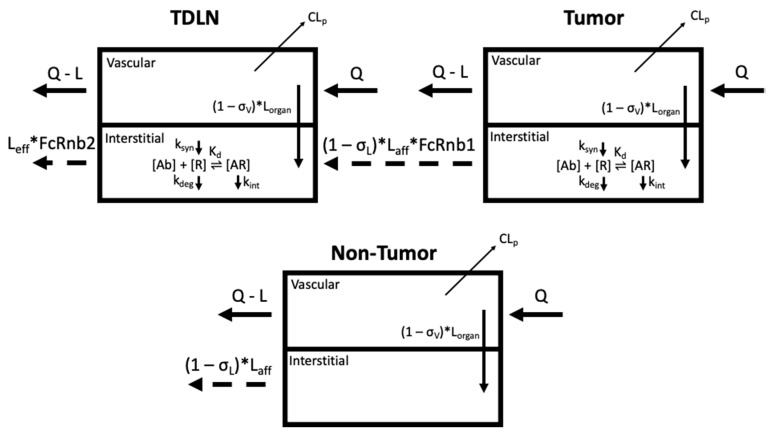
Schematic of the PBPK model to capture antibody distribution gradients from the organ-specific primary tumor (PT) to the tumor-draining lymph node (TDLN) (top). The non-tumor tissue compartment is a lumped compartment depicting antibody distribution in non-tumor sites of the system (bottom). CL_p_ denotes antibody clearance from the system. Antibody [Ab] engagement with its cognate receptor [R] is characterized in both the tumors and lymph nodes. The target [R] turnovers (biosynthesis (k_syn_) and degradation (k_deg_)) are both considered. The endocytosis of antibody–receptor complexes (k_int_) is also defined. The trafficking of antibodies from the tumors to the TDLNs are subject to nonspecific pinocytosis and FcRn-salvage in the lymphatic endothelial cells (FcRnb1). Other symbols are defined in Tables 1 and 2. Figure prepared in Biorender.

**Figure 2 antibodies-11-00028-f002:**
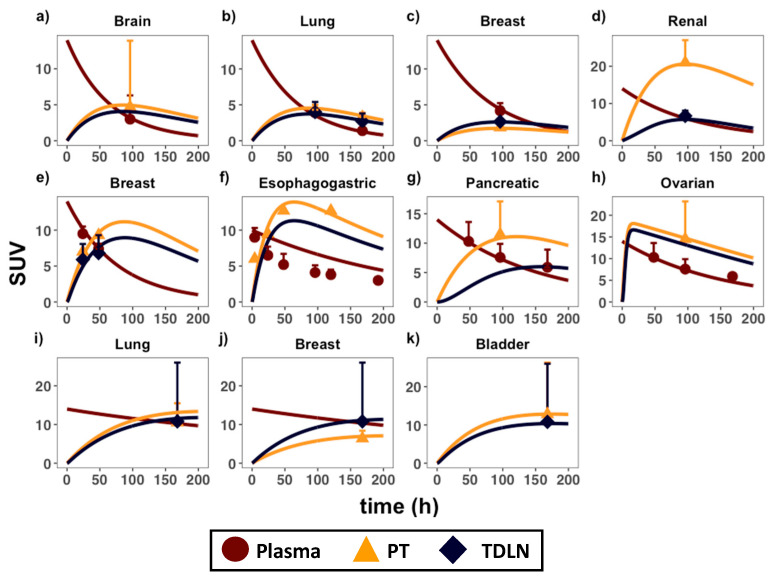
PBPK model calibration in anatomically distinctive tumors and TDLNs. Antibody distribution data in the plasma (circles), primary tumors (triangles), and TDLNs (diamonds) in eight immuno-PET biodistribution studies were applied to calibrate the model. Each calibrated plot is an organ-specific primary tumor, representing the following immunoPET-antibody pairs: (**a**) ^89^Zr-fresolimumab (glioblastoma) [16]; (**b**) ^89^Zr-bevacizumab (non-small-cell lung cancer; NSCLC) [17]; (**c**) ^89^Zr-bevacizumab (breast cancer) [18]; (**d**) ^89^Zr-bevacizumab (renal cell carcinoma) [19]; (**e**) ^64^Cu-DOTA-trastuzumab (breast cancer) [20]; (**f**) ^89^Zr-trastuzumab (esophagogastric cancer) [21]; (**g**) ^89^Zr-MMOT0530A (pancreatic cancer) [22]; (**h**) ^89^Zr-MMOT0530A (ovarian cancer) [22]; (**i**) ^89^Zr-atezolizumab (NSCLC) [23]; (**j**) ^89^Zr-atezolizumab (breast cancer) [23]; (**k**) ^89^Zr-atezolizumab (bladder cancer) [23]. SUV = standard uptake value. Data are presented as mean +/− standard deviation (SD). Please consult Appendix F for the derivation of the SUV parameter.

**Figure 3 antibodies-11-00028-f003:**
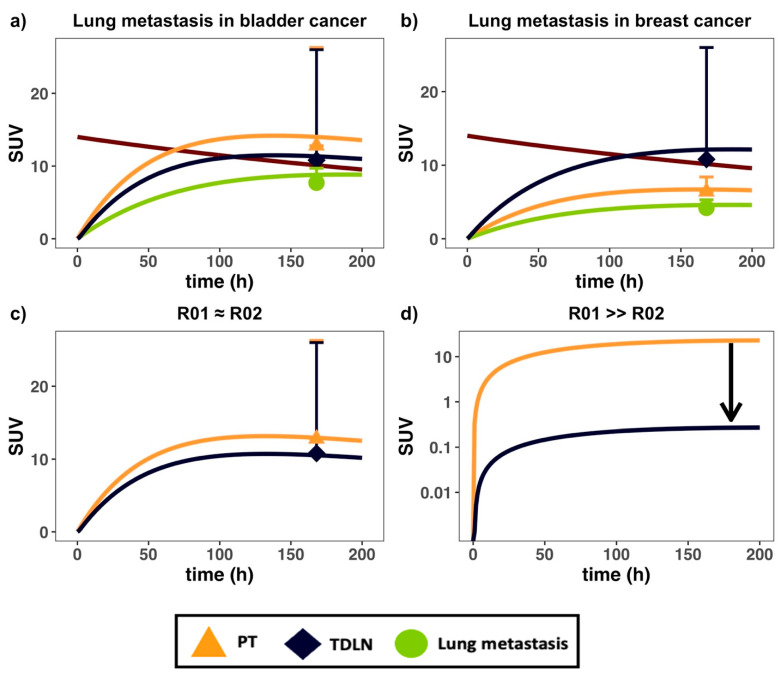
The PBPK model captured antibody distribution in lung metastases in patients with differing primary tumor types. Antibody PK data in (**a**) bladder and (**b**) breast cancer patients, both bearing lung metastases, were digitized and implemented into the model [23]; (**c**) effects of similar PT (red) and TDLN (green) target density (i.e., R01 = R02) on concentration gradients; (**d**) simulation of the effect of high PT target density relative to TDLN (i.e., R01 > R02) on gradient. The black arrow in (**d**) denotes the change in gradient owing to high PT metastatic target density. SUV = standard uptake value. Data are presented as mean ± SD.

**Figure 4 antibodies-11-00028-f004:**
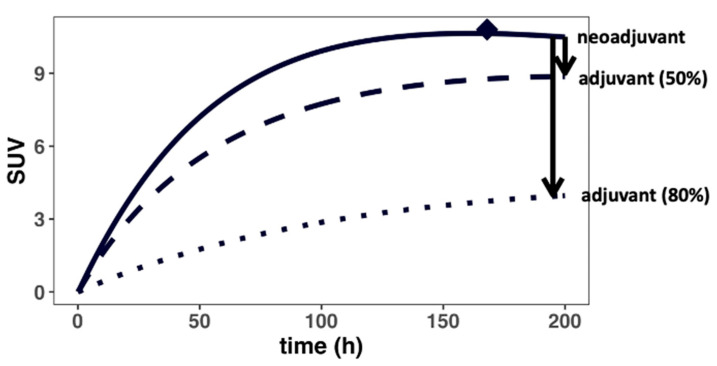
Surgical resection could drastically reduce antibody distribution into the TDLNs. Surgery (i.e., adjuvant) disrupts the local lymphatic vasculature, impairing lymph flow surrounding the tumor regions, resulting in a steeper concentration gradient (i.e., lower antibody delivery) for the residual TDLNs. The black arrows denote the change in gradient in cases where the lymph flows surrounding the residual TDLNs are reduced by 50 and 80% by the surgery, respectively.

**Figure 5 antibodies-11-00028-f005:**
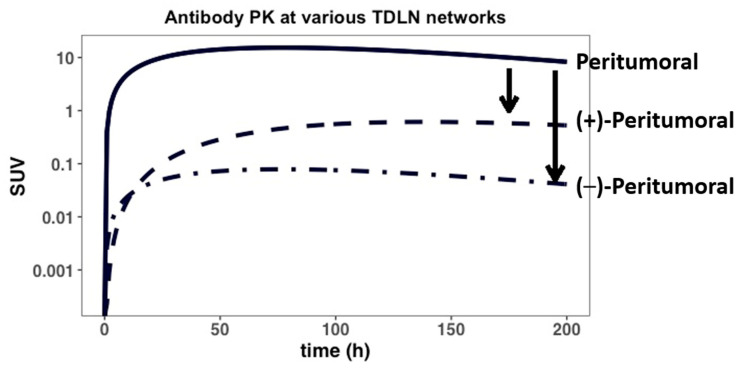
The antibody distribution between peritumoral and intra-tumoral TDLNs. Tumor-induced inflammation can impair, to different degrees, the lymphatic drainage to both tumor-positive (+) and tumor-negative (−) intratumoral lymphatic vessels/TDLNs, while also enhancing lymphatic drainage to peri-tumoral vessels and TDLNs. The model predicts the effect of PT-induced inflammation on antibody distribution gradients (i.e., black arrows) between the various TDLN networks.

**Figure 6 antibodies-11-00028-f006:**
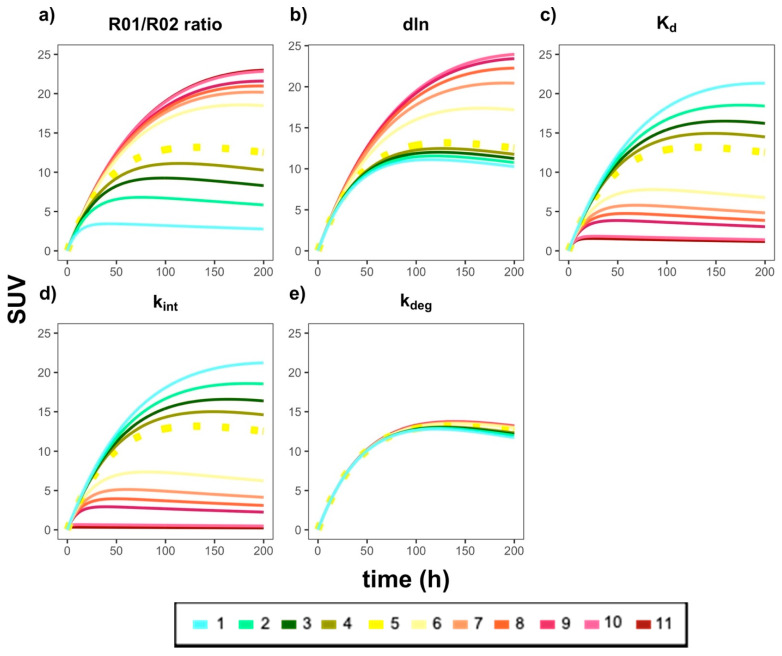
The effects of the key model parameters on antibody distribution profiles in the tumors and TDLNs. Each plot represents a specific model parameter that was adjusted by multiplying the calibrated value by the following fold changes: (1) 0.1×; (2) 0.3×; (3) 0.5×; (4) 0.7×; (5) 1.0×; (6) 3.0×; (7) 5.0×; (8) 7.0×; (9) 10.0×; (10) 50.0×; (11) 100.0×. The parameters that were adjusted, and their resulting predicted curves, were as follows: (**a**) R01/R02 ratio (i.e., ratio of tumor burden in the PT versus TDLN); (**b**) dln; (**c**) K_d_; (**d**) k_int_; (**e**) k_deg_. The dashed yellow line in each plot is the simulation for the calibrated (i.e., 1.0×, (5)) parameter value in the model. For best results, please view in color.

**Table 1 antibodies-11-00028-t001:** Individual table listing organ-specific parameters used to calibrate the model.

Primary Tumors	Antibody	σ_V_	L_organ_ ^a^ (L/h)	V_ISF,PT_ ^b^ (L)	V_p_ (L)
GBM [16]	^89^Zr-fresolimumab	0.94	0.05	0.265	5.0
NSCLC [17]	^89^Zr-bevacizumab	0.85	0.012	0.175	5.0
Breast [18]	^89^Zr-bevacizumab	0.95	0.008	0.112	5.0
Renal [19]	^89^Zr-bevacizumab	0.97	0.082	0.060	5.0
Breast [20]	^64^Cu-DOTA-trastuzumab	0.65	0.008	0.112	5.0
Esophagogastric [21]	^89^Zr-trastuzumab	0.95	0.007	0.005	7.0
Pancreatic [22]	^89^Zr-MMOT0530A	0.87	0.004	0.029	5.0
Ovarian [22]	^89^Zr-MMOT0530A	0.95	0.004	0.001	5.0
NSCLC [23]	^89^Zr-atezolizumab	0.80	0.012	0.175	5.0
Breast [23]	^89^Zr-atezolizumab	0.90	0.008	0.112	5.0
Bladder [23]	^89^Zr-atezolizumab	0.86	0.001	0.0084	5.0

^a^ L_organ_ (organ-specific lymph flow), 0.2% of organ blood flow [25,26]; ^b^ V_ISF,PT_ (PT interstitial fluid volume), 20% of total organ/tissue volume; σ_V_ (vascular reflection coefficient). Organ-specific parameters that were shared by all organ systems include: σ_L_ (lymphatic reflection coefficient) = 0.2 [27]; L_aff_ (afferent lymph flow) = 0.004 L/h [28]; L_eff_ (efferent lymph flow) = 0.004 L/h [28]; V_ISF, TDLN_ (TDLN interstitial fluid volume) = 0.0000584 L, 20% of volume of average LN [28]; [FcRn] (FcRn concentration in endothelial cells) = 40 uM [29].

**Table 2 antibodies-11-00028-t002:** Antibody and tumor-specific parameters in the model.

Primary Tumors	Antibody	R01 (nM)	R02 (nM)	K_d_ (nM)	K_d,FcRn_ (nM)	dln	CL_p_ (L/h)
GBM [16]	^89^Zr-fresolimumab	1	1	1.7	2400	19	0.075
NSCLC [17]	^89^Zr-bevacizumab	10	10	0.058	2400	21	0.07
Breast [18]	^89^Zr-bevacizumab	1	1.5	0.058	2400	20	0.06
Renal [19]	^89^Zr-bevacizumab	10	2.5	0.058	2400	21	0.042
Breast [20]	^64^Cu-DOTA-trastuzumab	100	100	5	774	61	0.063
Esophagogastric [21]	^89^Zr-trastuzumab	30	30	5	774	63	0.0288
Pancreatic [22]	^89^Zr-MMOT0530A	1000	1000	0.5	2400	21	0.033
Ovarian [22]	^89^Zr-MMOT0530A	24	24	0.5	2400	24	0.033
NSCLC [23]	^89^Zr-atezolizumab	7	7	0.43	2400	21	0.0083
Breast [23]	^89^Zr-atezolizumab	1	2.5	0.43	2400	20	0.0083
Bladder [23]	^89^Zr-atezolizumab	11	11	0.43	2400	24	0.0083

R01: baseline target concentration in PT [29]; R02: baseline target concentration in TDLN [29]; K_d_: antibody affinity to target [30,31,32]; K_d,FcRn_: antibody affinity to FcRn [33]; dln: shape factor [28,34]; CL_p_: antibody clearance from plasma [35,36,37,38]. Antibody/tumor specific parameters that were shared by all organ systems include: k_deg_: target degradation rate, 0.01 1/h; k_int_: antibody-target complex internalization rate, 0.01 1/h. Please consult Appendix B for derivation of dln parameter.

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
