# Peer review of "A Physiologically Based Pharmacokinetic Framework for Quantifying Antibody Distribution Gradients from Tumors to Tumor-Draining Lymph Nodes"

_2073-4468, 2022, doi:10.3390/antib11020028_

Round 1

Reviewer 1 Report

The authors presented a physiologically based pharmacokinetic framework for quantifying antibody distribution gradients from tumors to tumor-draining lymph nodes (TDLN). The primary tumor and TDLN signaling axis is very important for the efficacy of the immunotherapy. The activation of antitumor immunity is highly dependent on adequate therapeutic exposure into the primary tumor and TDLNs, but surgery might be resulted in less effective distribution of therapeutic antibodies into the tumor lesions and the surrounding TDLN, which causes decreased efficiency of the immunotherapy. The authors highlighted the importance of their model, as this could be used for the quantitative analysis and prediction of drug pharmacokinetics and tissue distributions.

For model building and calibration, antibody human immune-PET data of eleven different antibodies from eight immune-PET biodistribution studies were applied. These data were published by different independent researchers, whose work are cited in this manuscript.

The introduction, the importance of this study, the discussion, as well as the methods applied are clear, however the figures and tables are not easy to understand. In Figure 2 and Figure 3 the standard deviations are very high in some cases, and it is not clear why they are presented only in one direction. In addition, in Figure 1 a, c, d, i, j and k only one point is presented, while other figures show more than one points. This might come from the original study, but it is hard to understand how these curves were generated. Figure 2f shows only PT and plasma data, but contains three curves and it is not explained, what these curves are. It would be good to discuss, how relevant these data are, if the standard deviation is that high. The question is the same for Figure 3, where the standard deviations are high as well, there is only one point for each sample type and one more curve than dataset. Figure 6 shows the effect of key parameters on antibody distribution profiles in the tumors and TDLNs and the parameters were adjusted by multiplying the calibrated value by different fold changes. Where these fold changes come from and why are they different? Table 1 heading shows the reference [22]. Does this mean that the parameters presented came from this citation? If yes, why there are no citations for the other two parameters? In general, the results are interpreted clearly in the text, but not in figures and tables, which could be improved.

The manuscript contains 46 citations, some of them are published in the last few years, so the authors present up-to-date information, and the ratio of self-citations is optimal.

The methods applied are detailed and seems to be reproducible, the conclusions are consistent with the evidence presented.

To summarize, the manuscript describes a highly important model, which could be used for the prediction of drug pharmacokinetics and tissue distribution. The data and explanations in the text are clear, but the figures and tables are not easy to understand. I recommend the manuscript for publication in Antibodies after minor revision.

Author Response

Thank you for the insightful comments. We have included all response in the attached document. 

Reviewer 2 Report

The authors present a model tool to predict antibody distribution between tumor and tumor draining lymph node. To understand this scenario is important for immunotherapy.

Tumor targeting by antibody is affected by the expression of targets on tumor cells. The density of tumor epitopes controls the therapeutic potential. As primary tumors and metastatic tumor mutate the target density changes.

The author should consider this parameter in their model.

Author Response

Thank you for your kind comments and we have included our response in the attached document. Well regards. 
